

Natural Hazards and Earth System Sciences

Discussions

# Characteristics of ground motion and threshold values for colluvium slope displacement induced by heavy rainfall: a case study in northern Taiwan

**C.-J. Jeng and D.-Z. Sue**

Huafan University, New Taipei City, Taiwan

Received: 2 December 2015 – Accepted: 10 December 2015 – Published: 15 January 2016

Correspondence to: C.-J. Jeng (jcjhf@cc.hfu.edu.tw)

Published by Copernicus Publications on behalf of the European Geosciences Union.

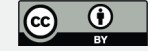

## Abstract

The Huafan University campus is located in the Ta-Lun Shan area in northern Taiwan, which is characterized by a dip-slope covered by colluvium soil of various depths. For slope disaster prevention, a monitoring system was constructed that consisted of

inclinometers, tiltmeters, crack gages, groundwater level observation wells, settlement and displacement observation marks, rebar strain gages, concrete strain gages, and rain gages. The monitoring data derived from hundreds of settlement and displacement observation marks were analyzed and compared with the displacement recorded by inclinometers. The analysis results revealed that the maximum settlement and

displacement were concentrated on the areas around the Hui-Tsui, Zhi-An, and Wu-Ming Buildings and coincided with periods of heavy rainfall. The computer program STABL was applied for slope stability analysis and modeling of slope failure. For prevention of slope instability, a drainage system and tieback anchors with additional stability measures were proposed to discharge excess groundwater following rainfall.

Finally, threshold value curves of rainfall based on slope displacement were proposed. The curves can be applied for predicting slope stability when typhoons are expected to bring heavy rainfall and should be significant in slope disaster prevention.

## 1   Introduction

Colluvium soils on slope surfaces in Taiwan result from geological fractures caused by

plate convergence and are generally characterized by looseness in composition, large porosity, high permeability and consequently, low stability. Colluvium soils are widely spread and gap-graded with a small sand fraction and exhibit a bimodal feature in grain size distribution (GSD) (Zhang and Chen, 2005; Li and Zhang, 2008).

Rainstorms frequently trigger colluvium landslides. For example, in November 1993,

more than 800 landslides were triggered by rainstorms on Lantau Island, Hong Kong (Dai and Lee, 2002). Fleming and Johnson (1994) studied colluvium landslides in

Discussion Paper | Discussion Paper | Discussion Paper | Discussion Paper |

**NHESSD**

doi:10.5194/nhess-2015-333

**Ground motion and threshold values for colluvium slope displacement**

C.-J. Jeng and D.-Z. Sue

Cincinnati, Ohio, and showed a significant difference in landslide behavior in slopes underlain by thick and thin colluvium. Corominas et al. (2002) discussed the landslides triggered by rainfall in Spain. An assessment of the relationship between rainfall and landslides on natural terrain from 1985 to 2000 was conducted by Evans (1997) and Ko (2003). The assessment showed that the occurrence of landslides on natural terrain increases exponentially with the normalized maximum 24 h rainfall. Furthermore, Li and Zhang (2008) discussed the stability of colluvium soil slopes during rainstorms. They measured hydraulic properties of colluvium soils with varying grain sizes and used them to analyze the stability of colluvium slopes during rainfall. In Taiwan, colluvium slope disasters related to rainfall are common (Jeng et al., 2007; Jeng and Lin, 2011; Pan et al., 2008). Jeng and Lin (2011) documented the variation of matrix suction of colluvium soil in different precipitation conditions with varying vegetation at the same site as this paper. In their study, both the in-situ monitoring results and laboratory tests for the undisturbed specimens taken from the field were analyzed. In addition, the relationship between shearing strength and matrix suction for the colluvium soil were also included.

The threshold values for disasters associated with typhoons have been studied extensively. Wang and Yeh (2011) analyzed the data of 265 automatic rain gage stations and 25 surface weather stations of the Central Weather Bureau of Taiwan as well as tracks of typhoons, and set the first value above the 99th percentile of each station as the threshold value of extreme hourly rainfall. The locations of extreme hourly rainfall and the distribution characteristics of its occurrence frequency over Taiwan were then revealed. Hu and Liao (2010) developed a model using Gaussian processes and defined a threshold rainfall factor as an index to estimate the likelihood of landslide occurrence due to rainfall. González and Mayorga (2004) proposed rainfall thresholds that triggered landslides in Colombia by analyzing relationships between the accumulative rainfall and the event rainfall. The country was then divided into four zones and a threshold curve was obtained for each zone. However, most studies

**NHESSD**

doi:10.5194/nhess-2015-333

**Ground motion and threshold values for colluvium slope displacement**

C.-J. Jeng and D.-Z. Sue

merely correlate rainfall records with the disaster event and seldom consider slope displacement for determining a threshold value.

This paper focuses on the effects of heavy rainfall on the colluvium slopes in northeastern Taiwan. The significance of the settlement and displacement of the slopes recorded by 295 monitoring marks for ground motion is evaluated, including the results from over thirty inclinometers in the study area. Finally, threshold value curves that consider the displacement of slopes due to typhoon rainfall are established.

## 2   Basic information of the field case

The field case discussed in this research is the Huafan University campus in northeastern Taiwan with global coordinates of 121°41′33.6″ E and 24°58′49.6″ N. (Fig. 1). The campus is located on a geological dip-slope toward the southwest within the Ta-Lun Shan area with an elevation ranging from 450 to 550 m (Fig. 1). For risk management and research on slope stability, a monitoring system was set up in 2001 and data was collected for over ten years. The monitoring system includes inclinometers, tiltmeters, crack gages, water level observation wells, settlement and displacement monitoring marks, rebar strain gages, concrete strain gages, and rain gages. The locations of inclinometers as well as settlement and displacement monitoring marks are shown in the figures included in this paper for ease of interpretation. According to the rain gage records from 2003 to 2010, the average annual rainfall was about 4000 mm, most of which was attributed to torrential rainfall concentrated during the typhoon seasons, and the maximum monthly rainfall recorded was 1208 mm and occurred in October 2010 (Fig. 2). The displacement of point G4 on the Hui-Tsui Building (B1 in Fig. 1) is also shown in Fig. 2 for correlation with displacement.

**NHESSD**

doi:10.5194/nhess-2015-333

**Ground motion and threshold values for colluvium slope displacement**

C.-J. Jeng and D.-Z. Sue

Discussion Paper | Discussion Paper | Discussion Paper | Discussion Paper |

**NHESSD**

doi:10.5194/nhess-2015-333

**Ground motion and threshold values for colluvium slope displacement**

C.-J. Jeng and D.-Z. Sue

# 3    Geological conditions

The bedrock (basement rock) of the study site belongs to the Miocene Mushan Formation. It consists of sandstone (SS) and thin alternating layers of sandstone and shale (SS-SH) and is overlain by 10–20 m thick colluvium. The attitude of the bedrock
has a strike in the east–west direction with a dip anchor from 10 to 20° toward the south (Fig. 1). Successive geological investigations in this area reveal two small-scale faults. The first fault, the Nanshihkeng Fault trending northeast found during the early period of the investigations, is a thrust with a fault plane dipping to the southeast. Meanwhile, the second fault (i.e., the A Fault) trends northwest and is an oblique strike-slip fault with
its southwest wall relatively throwing up, being truncated by the Nanshihkeng Fault.

Boring logs of boreholes with high-quality cores reveal that the bedrock near the faults is more fractured than the other areas. In addition, two dimensional (2-D)-resistivity image profiling (RIP, with positions as shown in Fig. 3) performed thrice in 2007 is used to investigate the dimension and attitude of the shear zone, and
the distribution and depth of the bedding slips. Results of the 2-D image profiles (Fig. 4) depict an apparent change in the groundwater body between the three stages of measurements and also indicate the position of the A Fault with a high dipping angle at a distance of 63–75 m along RIP-1, which coincides with that shown in the geological map (Fig. 1). In addition, the increase in groundwater within zones A and
B (Fig. 4) may reflect a geologically fractured structure. Careful examination of the SI-1 cores (position shown in Fig. 3) reveals that most of the cores at depths of 34.9–60 m are heavily fractured with gouge and breccia intercalated, and the dip angles of the bedding vary chaotically (Fig. 5). In Figs. 1 and 5, SS-SH-1 and SS-SH-2 denote the first and second layer of sandstone interbedded shale, respectively, while SS-1
and SS-2 represent the first and second layer of sandstone, respectively. These facts indicate that a thick-bedded bedding slip to the north of the A Fault lies below the main campus. The thick sandstone unit encountered between 51.3 and 60 m may be equivalent to the rock unit SS-2 that contributes to the confined aquifer, so that the SI-1

Discussion Paper | Discussion Paper | Discussion Paper | Discussion Paper

borehole has an eternal piezometric level always greater than the ground surface as observed by inserting an extension pipe into the borehole. The fracture zones, failure planes, and bedding slip are interpreted directly by observing the cores. Borehole SI-1 was drilled through SS-SH-2 to SS-2, which is sandstone belonging to a confined aquifer, leading to the squeezing up of groundwater with a higher table than the ground surface. Two tentatively proposed interpretative hydrogeological maps shown in Figs. 3 and 5 are synthesized from borehole core logs, geological cross-sections, water level observations, and geophysical 2-D-resisitivity profiles. The groundwater direction shown in Fig. 3 is identified by the results of the 2-D-resisitivity profiles and is also affected by the intersection of the two faults.

It is important to point out that the south corner of the Wu-Ming Building (B2) is not only located near the intersection of the two faults mentioned above, but also at the corner with the lowest elevation among the main campus buildings. Since groundwater tends to flow toward this corner, the basement foundation of the Wu-Ming Building (B2) has undergone tremendous compressive stresses arising from the entire upper slope. Furthermore, two groundwater regimes of different properties may be distinguished (Fig. 5). One is unconfined under the main building area, which is supposed to develop in thick surface colluvium, artificial fills, and parts of underlying fractured bedrock, while the other is the deep-seated confined groundwater regime of the deep unit SS-2 aquifer.

## 4 Ground motion monitoring results

### 4.1 Settlement and displacement monitoring marks

Since 2001, hundreds of settlement and displacement monitoring marks have been set up and recorded every six months. Some additional marks were gradually included over the years. The data were recorded until January 2011 and a total of 295 marks were collected within this period as shown in Fig. 6. The investigations were performed by

**NHESSD**

doi:10.5194/nhess-2015-333

**Ground motion and threshold values for colluvium slope displacement**

C.-J. Jeng and D.-Z. Sue

a theodolite (TOPCON GTS-301) and a level (AE-5). The accuracy of the investigation was controlled to be within 1/5000 by a plane triangulation method with Global Positioning System, or GPS, (TRIMBLE 4800) measurements from six fixed station points (Fig. 6). The six fixed station reference points for measurements are also shown in Fig. 6. Closed guided lines are obtained by derived surveying proceeding from those reference points, and all points of displacement and settlement are then acquired. Four traverse points were then laid within the survey area and rechecked to maintain the accuracy to within 1/10000. The coordinates and the elevation of each observation mark were then surveyed based on these traverse points. The displacement and settlement values were obtained by comparing the coordinates and elevation results of each survey to the initial results from the first survey. The data marks were then divided into two categories, namely one for buildings (denoted by blue triangles in Fig. 6) and the other for roads and land (denoted by pink circles in Fig. 6).

## 4.2 Results of settlement and displacement distribution

Figure 6 shows ground settlement measured by the 295 monitoring points within the campus. The values of settlement or heaving are divided into five levels from 5 to 25 mm and are represented by different size circles. As indicated in Fig. 6, the maximum annual settlement is greater than 20 mm and is distributed around the Hui-Tsui Building (B1), Wu-Ming Building (B2), and Zhi-An Building (B3). Although these values do not reach a level of danger based on general management criteria, for some areas, the cumulative settlement exceeds 10 cm as revealed by the 10 year data. Moreover, comparing the settlement distribution in Fig. 6 to the thickness of the filled land shown in Fig. 7, the settlement is obvious in filled land areas of greater thickness such as the Asoka Square (B6), Sports Ground (B8), and Basketball Court. It is speculated that these areas with higher settlement must be related to the thicker fill. For a few heaving points such as the upper slope of the Asoka Square (B6), it appears to be the consequence of the surface concrete pavement heaving caused by an extrusion of the slope slide, but not by the ground surface conditions.

**NHESSD**

doi:10.5194/nhess-2015-333

**Ground motion and threshold values for colluvium slope displacement**

C.-J. Jeng and D.-Z. Sue



Figure 8 shows the displacement tracks of each observation mark on the plan map. It can be seen that the main displacement direction is down to the slope in the southwest direction. In addition, the larger displacement distributes, same as the aforementioned settlement distribution, around the areas of Hui-Tsui Building (B1), Wu-Ming Building (B2), and Zhi-An Building (B3). However, the upper slope of the Asoka Square (B6) shows movement in the upward direction of the slope. This can also be attributed to the aforementioned heave phenomenon caused by extrusion of the slope slide, resulting in an overturning condition to occur in that area. The cracks that developed in the crown area appear to be consistent with this analysis. Additionally, the movement direction of the slope at the Sports Ground (B8) is different from that of the upper slope at the Dormitory (B4) building since they are departing into different sliding blocks.

### 4.3 Correlation between settlement, displacement, and rainfall

According to the results of the aforementioned settlement and displacement distributions, the most critical areas are concentrated on the Hui-Tsui Building (B1), Wu-Ming Building (B2), and Zhi-An Building (B3). As expected, the most significant contributing factor is rainfall. The correlation between settlement, displacement, and rainfall for the Hui-Tsui Building (B1) is discussed below.

The tendency of displacement with respect to time shown in Fig. 9 is generally similar to that of settlement vs. time shown in Fig. 10. Based on the increment tendency, they may both be separated into four distinct time segments: (1) May 2001 to March 2002, (2) September 2004 to June 2006, (3) June 2007 to April 2009, and (4) April 2010 to January 2011.

By comparing the displacement and settlement time in Figs. 9 and 10 with the rainfall record in Fig. 2, the displacement and settlement of the slope appear to have a strong correlation with the rainfall record. For example, one settlement point (point G4 in Fig. 6) at the Hui-Tsui Building (B1) shown in Fig. 2 indicates its strong correlation with the monthly rainfall record. In addition to the rainfall, another important factor in the displacement and settlement of the slope is construction. For instance, within the third

Discussion Paper | Discussion Paper | Discussion Paper | Discussion Paper | Discussion Paper |

**NHESSD**

doi:10.5194/nhess-2015-333

**Ground motion and threshold values for colluvium slope displacement**

C.-J. Jeng and D.-Z. Sue

time segment, a new Library and Information (L and I) Building (B5) was built with an excavation for the foundation at the toe of the slope. This may explain why the increased amount of displacement and settlement in the third time segment is greater than in the second time segment, despite the greater total segment of cumulative rainfall (sum of total rainfall for each time segment) in the second time segment (5550 mm) than the third time segment (4600 mm).

In addition to the cumulative amount of rainfall, intensity and duration also contribute to the threshold value that triggers displacement and settlement of the slope. By analyzing the relationship between displacement variation and monthly cumulative rainfall (Fig. 11), the warning value is reached when the monthly rainfall is equal to or greater than 500 mm (semi-accurate slide), and the action value is reached when the monthly rainfall is equal to or greater than 855 mm (accurate slide). It demonstrates that a rainfall threshold value of 855 mm per month is able to trigger a slide and settlement of the slope.

## 4.4 Inclinometer monitoring

Many inclinometers have been gradually installed since 2000 in the test field. Meanwhile, some of them were damaged due to deformation and settlement. At present, 32 boreholes remain functional and are recording measurement results. Displacement with depth recorded by the inclinometers (e.g., borehole No. SIS-14, Fig. 12) is presented and a sliding layer that deforms 10 to 11 m in depth is revealed. By analyzing the core of borehole No. SIS-14 (Fig. 13), sliding is found to occur within the fracture layer.

## 5 Relationship between inclinometer displacement and daily rainfall

Based on the monitoring results obtained in 2007, the relationship between displacement recorded by the inclinometer (No. SIS-11A) and daily rainfall

Discussion Paper | Discussion Paper | Discussion Paper | Discussion Paper

**NHESSD**

doi:10.5194/nhess-2015-333

**Ground motion and threshold values for colluvium slope displacement**

C.-J. Jeng and D.-Z. Sue

(Fig. 14) reveals that the effect of concentrated rainfall caused by Typhoon Krosa has significantly accelerated the displacement increment. For safety precaution, a preliminary relationship between displacement and daily rainfall for the campus derived from the data of more than thirty inclinometers is presented in Table 1. For daily rainfalls greater than 50 mm, a low bound movement of the slope with a corresponding displacement of about 0.5–2.0 mm will occur. During the periods when the daily rainfall exceeds 150 mm, a corresponding displacement between 4 and 6 mm will occur. When the daily rainfall is greater than 300 mm, a corresponding displacement of between 6 and 10 mm will take place.

## 6 Comparison of the results between displacement monitoring marks and the inclinometers

The displacement monitoring marks are able to characterize the ground surface deformation and the inclinometer pipes can describe the ground deformation for the entire depth. To investigate the deformation of the slope, the top of the inclinometer pipes was superimposed with the displacement monitoring marks. The comparison results, also shown in Fig. 8, indicate that the tendency of slope surface deformation obtained from both these data sets is generally consistent. The primary slope deformation is toward the southwest and south directions similar to that of the slope surface, which indicates that the slope soil generally moves downward along the slope.

## 7 Slope stability analysis

Besides the significant amount of slope deformation, there is concern regarding the integrity of the slope and any potentially hazardous contributing factors. To determine the slope stability, the limit equilibrium program STABL was applied for analysis. Since the study site consists of colluvium soil and a highly weathered rock and fracture layer, both circular failure and block failure were used for the analysis. The sliding depth

**NHESSD**

doi:10.5194/nhess-2015-333

**Ground motion and threshold values for colluvium slope displacement**

C.-J. Jeng and D.-Z. Sue

**NHESSD**

doi:10.5194/nhess-2015-333

**Ground motion and threshold values for colluvium slope displacement**

C.-J. Jeng and D.-Z. Sue

obtained via inclinometer monitoring was also investigated. Consequently, a simplified Bishop method with block failure was selected for the study. The input soil parameters shown in Table 2 are based on the average laboratory direct shear test results with core samples obtained when the inclinometer boreholes were drilled. The test specimens were acquired from the cores when the boreholes were drilled to mount inclinometers shown in Fig. 13. One of the specimens was pressed into a sampling ring for a direct shear test, and then trimming of both ends of the specimen was carried out to fit it to the sampling ring. The specimen was then tested in a direct shear test apparatus to obtain the shearing strength parameter $C$ and friction angle $\phi$ derived from failure envelope of the test. The parameters were then checked using a back analysis from the potential sliding surfaces observed from the inclinometers (Fig. 15). The cohesion and friction angle of the colluvium soil adopted were $C = 18.5$ kPa and $\phi = 29.6°$, respectively. The strength of the bedrock appeared to be at an intermediate condition between peak strength and residual strength when the slope was affected by slope movements over the last several years. Areas with similar groundwater variations and rainfall conditions were compared and the crack distribution on the slope was used to assess potential failure areas.

The sliding plan and failure patterns are the main parameters that STABL considers to decide sliding blocks. STABL first searches ten potential failure plans automatically with a minimum safety factor. Hereafter, according to observation results of the inclinometers and the trend of the dip-slope planar slide, the "BLOCK" failure pattern is chosen. To determine the groundwater table in normal and storm conditions in STABL, observation results of normal and typhoon storms conditions were adopted, respectively. As a result, the safety factor was 2.17 (Fig. 16), which is greater than the suggested value 1.5. However, for rainfall conditions when the groundwater table rises 3 m, the safety factor decreased to 0.86 and an unstable condition could then occur (Fig. 17). By comparing the block potential sliding surfaces in these two figures, the major difference appears to be the rising of the groundwater table to the sliding surface (Fig. 17). Thus, slope stability is significantly affected by the rise in groundwater during

rainfall events. On the other hand, the effects of short-term concentrated rainfall on the slope displacement are more detrimental than small amounts of long-term rain (Fig. 14). Thus, we conclude that the most important trigger factor of slope sliding is heavy rainfall caused by a typhoon. For example, Typhoon Krosa induced a slope displacement of approximately 20 mm which is at least five times greater than that caused by a regular rainfall (Fig. 14).

## 8 Review of sliding blocks based on settlement and displacement results

According to the inclinometer monitoring results over the years and surface cracks observed (Fig. 8), six sliding blocks in the slope within the campus may exist, and each sliding block has a displacement rate and a sliding direction differing from others (Fig. 18). The sliding blocks are defined on the basis of correlation between the positions of inclinometers and varying displacements with depth measured by the inclinometers. The depth contour of the groundwater table derived from drilling is used to interpret flow direction (Fig. 19). The groundwater flow paths are derived from the groundwater table contour, results of RIP methods, and distribution of curved morphology in topography. In addition, the Nanshihkeng Fault plays a role as an obstruction causing separation of groundwater flow.

Based on the observation results from ground surface marks, this study also evaluates ground surface movement, displacement direction, ground settlement (Fig. 6), and ground surface cracks (Fig. 8). A comprehensive evaluation of the monitoring results on the sliding blocks is then performed. The distribution of sliding blocks is reviewed and shown in Fig. 20, in which new findings support the presence of two new sliding blocks (blue curved lines). Block A-1 is located in the area around the Wu-Ming Building (B2), Asoka Square (B6), and Chea-Chau Building (B7), while block A-2 is located in the area around the Sports Ground (B8) where there is 20 m thick of fill as shown in Fig. 7. It can be seen that block A-2 coincides with block L1 and block A-1 is within block R1 (Fig. 20). This implies that the results of both the inclinometers

Discussion Paper | Discussion Paper | Discussion Paper | Discussion Paper | Discussion Paper |

**NHESSD**

doi:10.5194/nhess-2015-333

**Ground motion and threshold values for colluvium slope displacement**

C.-J. Jeng and D.-Z. Sue

**NHESSD**

doi:10.5194/nhess-2015-333

**Ground motion and threshold values for colluvium slope displacement**

C.-J. Jeng and D.-Z. Sue

and ground surface marks reflect similar trends in ground movement. Furthermore, movements of both blocks are revealed to occur in the shallow layer and are located in the active sliding areas. Numerous cracks (cf. Fig. 8) can be found along the ground surface in these areas. Moreover, the displacement tracks of the ground surface marks (Fig. 8) exhibit a trend similar to that of the displacement direction of the inclinometers. Based on the observations as well as comparisons with the settlement distribution in Fig. 6, block R1 is further subdivided into two small sliding blocks (i.e., sliding blocks A-3 and A-4).

## 9   Stabilization measures and the threshold value of rainfall

According to the aforementioned stability analysis results, the rise in groundwater during rainstorm conditions significantly impacts slope stability. Consequently, slope stability can be improved with the addition of drainage and drawdown systems and retaining structures. With budget and effectiveness consideration, the first step should focus on the area around the Wu-Ming Building (B2), where the groundwater and geological conditions are the least favorable. A detailed description of the improvement steps includes the following:

1. Improvement of the ground surface drainage system for water runoff. This can simply reduce infiltration but does not have an effect on lowering the groundwater table.

2. Installation of six catchpits with the horizontal drainage pipes (Fig. 21). This is expected to effectively draw down the groundwater level and should be performed first. However, due to a high construction cost, two of the six catchpits will be constructed first and their effectiveness will be evaluated via the monitoring system. The remaining catchpits will be installed based on the evaluation results from the first two catchpits.

3. Filling of ground surface cracks to prevent seepage of water runoff into cracks. This can only be viewed as a temporary measure until the entire slope is completely stabilized.

4. Construction of bored piles and tiebacks with ground anchors behind the Wu-Ming Building (B2) to strengthen the retaining structure at the toe of the slope (Fig. 22). These must be installed after the groundwater has been effectively drawn down.

Finally, for the safety management of the slope, a threshold curve (Fig. 23) is established to illustrate the relationship between rainfall intensity and accumulation and the observed slope displacement. This curve is derived from the rainfall records of numerous typhoon events over the past ten years with the corresponding slope displacement increment recorded by the ground surface marks. Threshold values of slope displacement for different sliding stages recommended by the Japan Association for Slope Disaster Management (JASDiM) are used to define three ranges (dashed lines in Fig. 23). According to the JASDiM, when a displacement rate of 2 mm per month of a slope occurs, the semi-accurate slide stage is reached; meanwhile, when a displacement rate of 10 mm per month occurs, the accurate slide stage is reached. In this study, we use these two stages to define the ranges of dangerous, warning, and safe values of typhoon cumulative rainfall and average intensity. In Fig. 23, the data points of slope displacement caused by typhoons of less than 2 mm (square marks) based on rainfall intensity and accumulation are separated from those exceeding 2 mm (diamond marks). Based on the displacement induced by these typhoons, three ranges (safe, warning, and dangerous) of rainfall threshold values are concluded and then recommended in this study. The general limits of cumulative rainfall for evacuation and day off announced by the government during typhoons are also presented in Fig. 23. These curves can be used for predicting slope stability when a typhoon is expected and heavy rainfall is predicted. Such information correlating rainfall to slope stability improves slope management and risk assessment. The proposed approaches are believed to be beneficial to minimize the risk of slope disasters.

**NHESSD**

doi:10.5194/nhess-2015-333

**Ground motion and threshold values for colluvium slope displacement**

C.-J. Jeng and D.-Z. Sue



# 10   Conclusions

This paper discussed the displacement and settlement of a slope, evaluated the sliding block theory, and analyzed slope stability. Summarization of the results allowed the establishment of stabilization measures and rainfall threshold values. Based on the results, the following conclusions can be made:

1. The areas with the most significant settlement and displacement are located around the Hui-Tsui Building (B1), Wu-Ming Building (B2), and Zhi-An Building (B3). Due to the lack of slope stability, surface cracks appear and several sliding surfaces are observed.

2. The deformation of the top of inclinometer pipes is consistent with the displacement monitoring marks. The findings indicate that the primary slope displacement is toward the southwest and south directions. In addition, the most important trigger factor for slope sliding is heavy rainfall caused by typhoons. The effect of concentrated rainfall caused by typhoons has accelerated the increment of slope displacement. A preliminary relationship between the displacement recorded by the inclinometers and daily rainfall within the campus is presented (Table 1).

3. The results of the slope stability analyses show that a rise in the groundwater table caused by typhoons is the most critical factor in slope stability. Therefore, a countermeasure of a drawdown groundwater table should be an effective method for slope stabilization.

4. The distribution of potential sliding blocks is evaluated by using slope displacement, settlement data, and the location of surface cracks. The sliding direction is strongly associated with the direction of groundwater flow. Depth or thickness of land filled also contributes to slope sliding.

## NHESSD

doi:10.5194/nhess-2015-333

**Ground motion and threshold values for colluvium slope displacement**

C.-J. Jeng and D.-Z. Sue

Discussion Paper | Discussion Paper | Discussion Paper | Discussion Paper

5. Several stabilization measures including catchpits with horizontal drainage pipes, bore piles, and tieback ground anchors to improve slope stability are recommended.

6. The threshold value curves of slope displacement based on rainfall intensity and cumulative rainfall are established. These curves can be used for predicting slope stability when typhoons are expected to bring heavy rainfall. The information presented in this study is expected to be of importance for slope disaster prevention.

**Author disclosure statement**

No competing financial interests exist.

*Acknowledgements.* The authors would like to thank the Ministry of Science and Technology of the Republic of China (No. NSC 102-2632-E-211-001-MY3) for financially supporting this research.

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

**Table 1.** Preliminary relationship between displacement of the inclinometer and daily rainfall for the campus.

| Stages | Cumulative daily rainfall $(\text{mm day}^{-1})$ | Assessment displacement of slope (mm) |
|--------|--------------------------------------------------|---------------------------------------|
| 1      | 30–50                                            | 0.5–2.0                               |
| 2      | 50–150                                           | 2.0–4.0                               |
| 3      | 150–300                                          | 4.0–6.0                               |
| 4      | 300–500                                          | 6.0–10.0                              |
| 5      | > 500                                            | > 10.0                                |

**NHESSD**

doi:10.5194/nhess-2015-333

**Ground motion and threshold values for colluvium slope displacement**

C.-J. Jeng and D.-Z. Sue

**Table 2.** Soil and rock parameters.

| Layer | Cohesion (kPa) | Friction angle (°) | Unit weight (kN m$^{-3}$) |
|---|---|---|---|
| Colluvium | 18.5 | 29.6 | 19.31 |
| Sandstone and shale | 41.8 | 32.1 | 25.52 |
| Sandstone | 38.7 | 32.7 | 23.86 |
| Fracture zone | 0 | 22.6 | 22.60 |

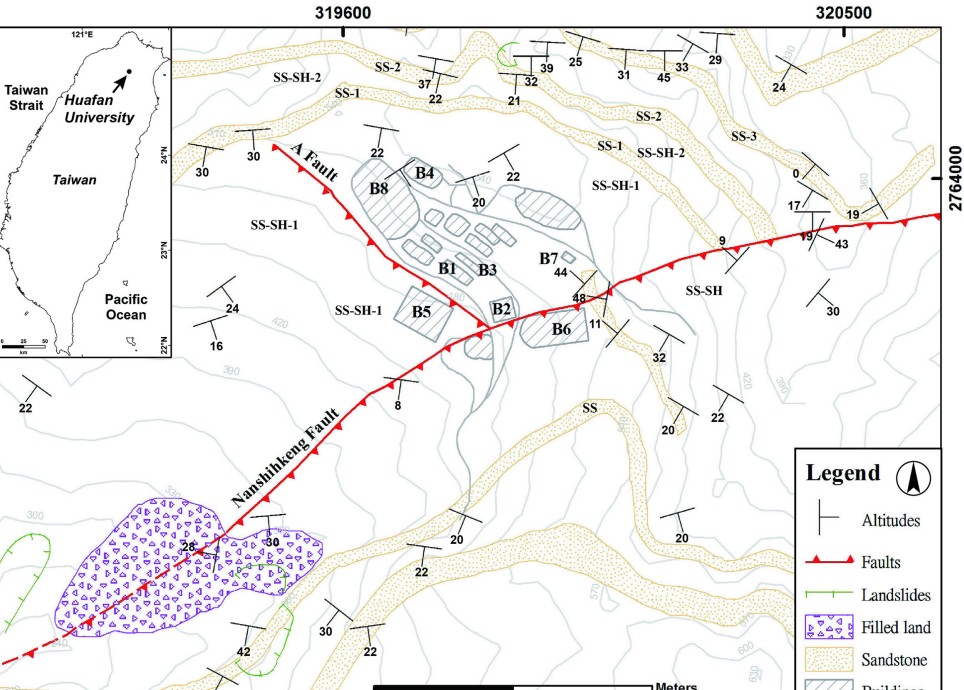

**Figure 1.** Field case (Huafan University campus) location and geology of the study area. According to the geological investigations, not only a dip-slope, but also two new faults (i.e., Nanshihkeng Fault and A Fault) are found within the campus. Building symbols are indicated as follows: B1: Hui-Tsui Building, B2: Wu-Ming Building, B3: Zhi-An Building, B4: Dormitory, B5: Library and Information (L and I) Building, B6: Asoka Square, B7: Chea-Chau Building, and B8: Sports Ground.

**NHESSD**

doi:10.5194/nhess-2015-333

**Ground motion and threshold values for colluvium slope displacement**

C.-J. Jeng and D.-Z. Sue

**NHESSD**

doi:10.5194/nhess-2015-333

**Ground motion and threshold values for colluvium slope displacement**

C.-J. Jeng and D.-Z. Sue

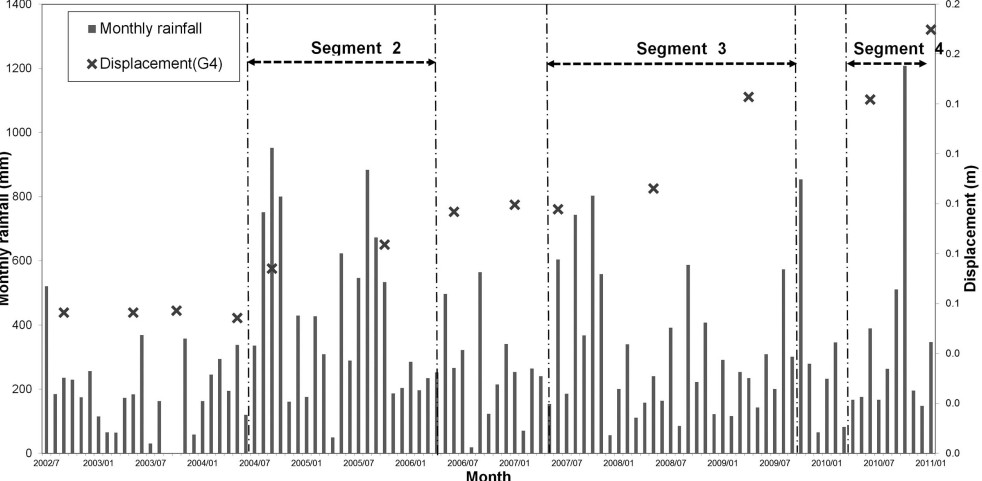

**Figure 2.** Correlation of monthly rainfall and slope displacement of point G4, which is an observation point at the Hui-Tsui Building (B1). It is enlarged and shown in Fig. 6. The slope displacement recorded at point G4 is measured to be greater during the periods of higher precipitation.

# NHESSD

doi:10.5194/nhess-2015-333

**Ground motion and threshold values for colluvium slope displacement**

C.-J. Jeng and D.-Z. Sue

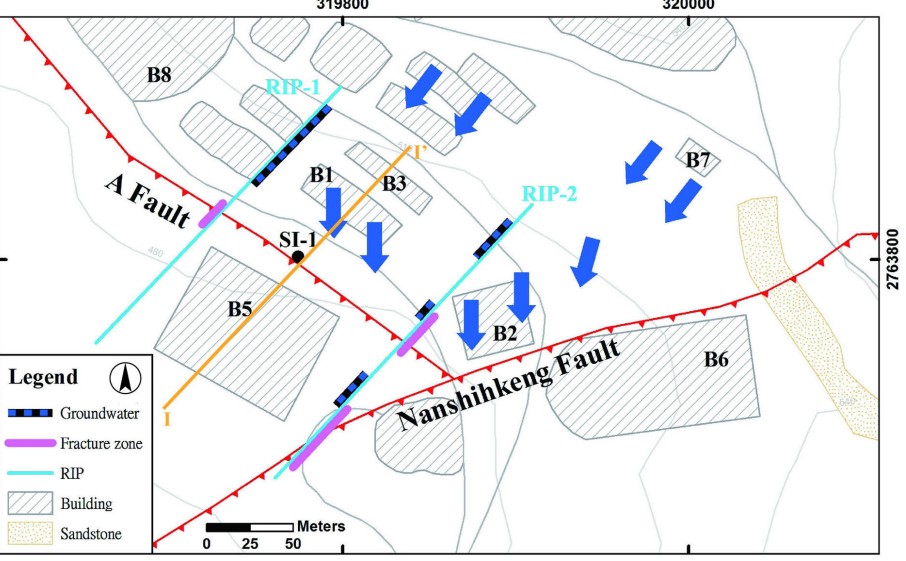

**Figure 3.** Hydrogeological map. Two 2-D RIP across the fault zone (i.e., A Fault) were carried out parallel to each other. The blue arrows indicate groundwater flow directions and the I-I' line represents a hydrogeological profile and is shown in Fig. 5.

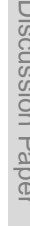

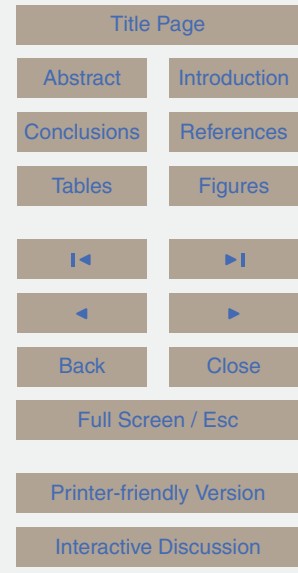

**Figure 4.** Image profile of RIP-1. The results (percentage change between the first and second measurements) reveal that the position of the groundwater body coincides with that of the fault zone of the A Fault, indicating that the groundwater flows along the fault zone. It also reveals an increase in groundwater in zones A, B, C, and a decrease in zone D.

Discussion Paper | Discussion Paper | Discussion Paper | Discussion Paper

**NHESSD**

doi:10.5194/nhess-2015-333

**Ground motion and threshold values for colluvium slope displacement**

C.-J. Jeng and D.-Z. Sue

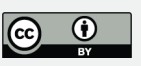

# NHESSD

doi:10.5194/nhess-2015-333

**Ground motion and threshold values for colluvium slope displacement**

C.-J. Jeng and D.-Z. Sue

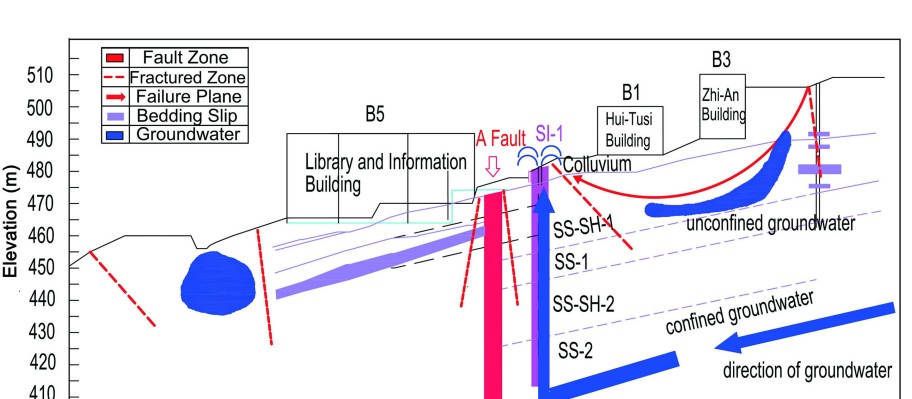

**Figure 5.** Hydrogeological profile I-I'.

Discussion Paper | Discussion Paper | Discussion Paper | Discussion Paper

## NHESSD

doi:10.5194/nhess-2015-333

**Ground motion and threshold values for colluvium slope displacement**

C.-J. Jeng and D.-Z. Sue

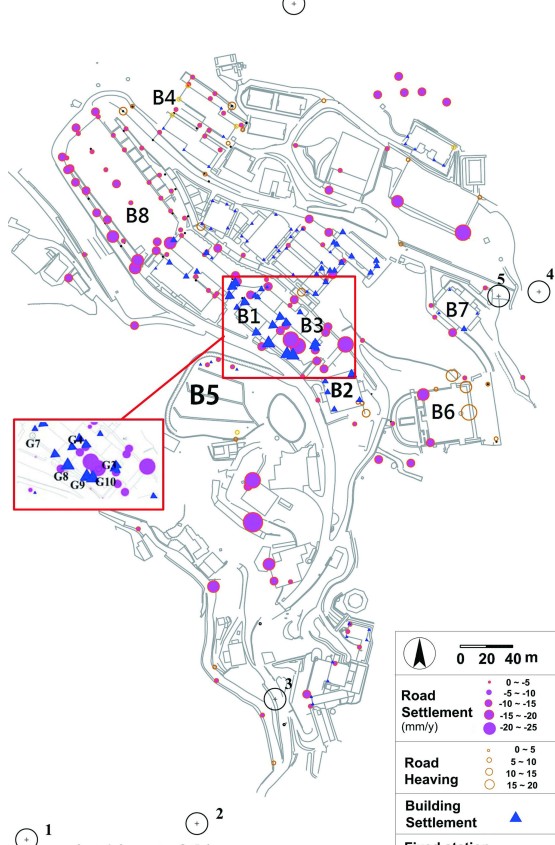

**Figure 6.** Ground settlement distribution within the campus. The settlement marks around buildings are denoted by solid triangles, the settlement marks for roads and other land areas are represented by solid circles, and the heaving points are indicated by hollow circles.

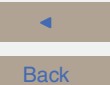

**Figure 7.** Distribution of thickness of filled land within the campus.

# NHESSD

doi:10.5194/nhess-2015-333

**Ground motion and threshold values for colluvium slope displacement**

C.-J. Jeng and D.-Z. Sue

Discussion Paper | Discussion Paper | Discussion Paper | Discussion Paper

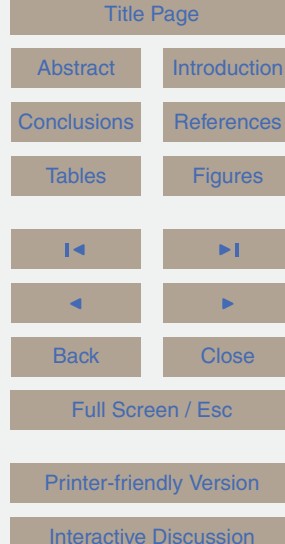

**NHESSD**

doi:10.5194/nhess-2015-333

**Ground motion and threshold values for colluvium slope displacement**

C.-J. Jeng and D.-Z. Sue



**Figure 8.** Displacement tracks and ground surface crack distribution. The scale of displacement has been enlarged 200 times to highlight its tendency.

**NHESSD**

doi:10.5194/nhess-2015-333

**Ground motion and threshold values for colluvium slope displacement**

C.-J. Jeng and D.-Z. Sue

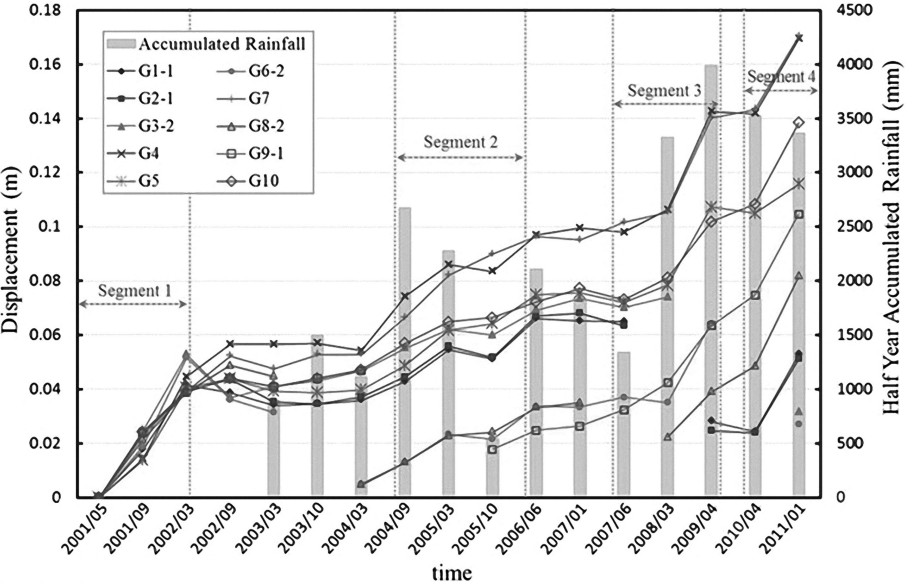

**Figure 9.** Curves of displacement and rainfall per half year for the Hui-Tsui Building (B1).



**NHESSD**

doi:10.5194/nhess-2015-333

**Ground motion and threshold values for colluvium slope displacement**

C.-J. Jeng and D.-Z. Sue

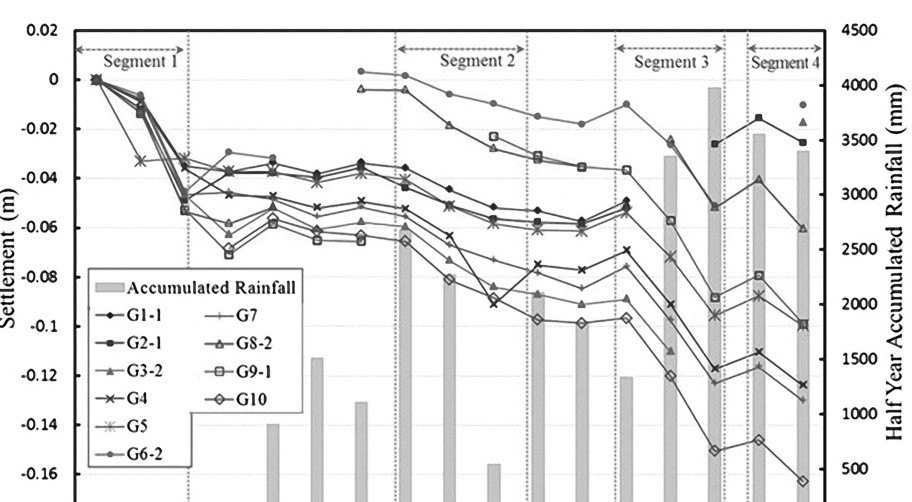

**Figure 10.** Curves of settlement and rainfall per half year for the Hui-Tsui Building (B1).



**NHESSD**

doi:10.5194/nhess-2015-333

C.-J. Jeng and D.-Z. Sue

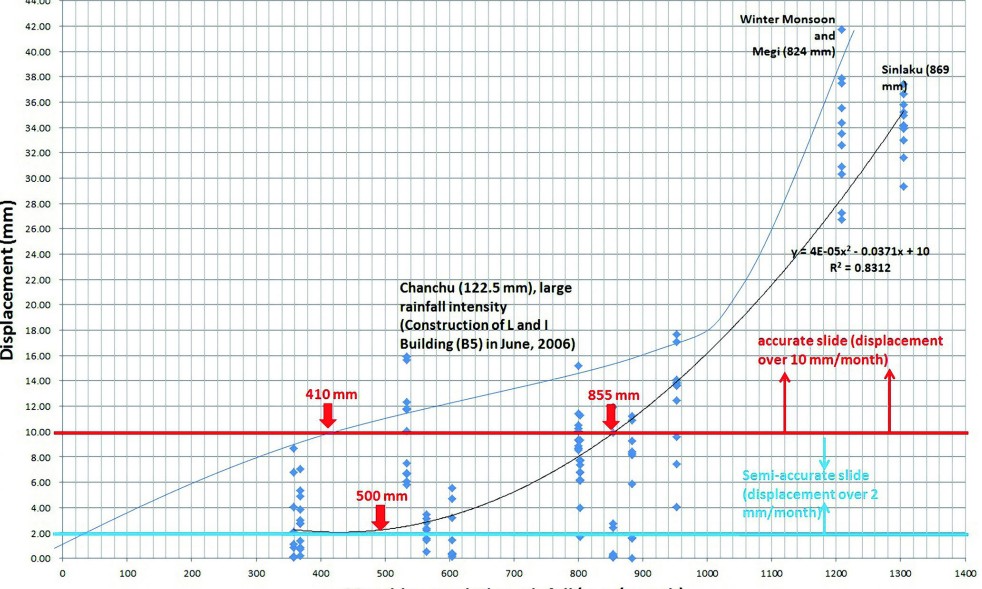

**Figure 11.** Relationship between monthly cumulative rainfall and displacement. The relationship curves are derived from a larger displacement variation of a fixed station point vs. monthly cumulative rainfall. By applying a binomial regression model, the black curve can be obtained to be the optimum relationship with a $R^2$ value of 0.831.

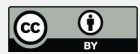

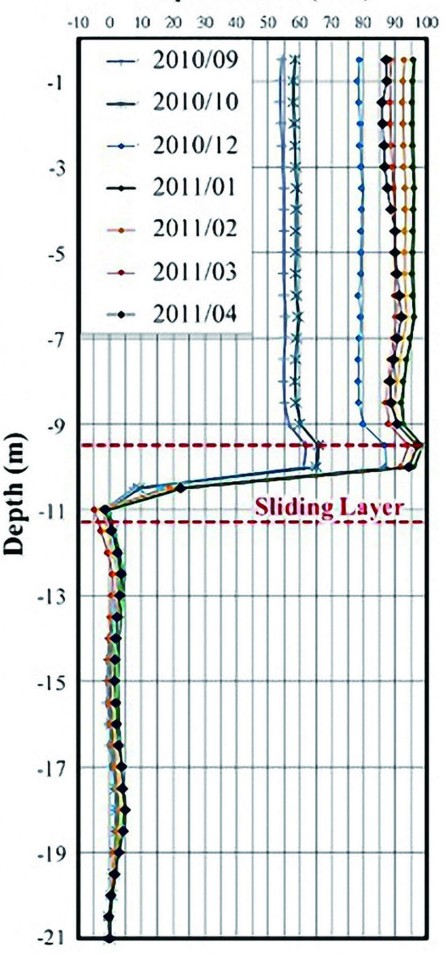

**Figure 12.** Displacement recorded by the inclinometer No. SIS-14.



**NHESSD**

doi:10.5194/nhess-2015-333

**Ground motion and threshold values for colluvium slope displacement**

C.-J. Jeng and D.-Z. Sue

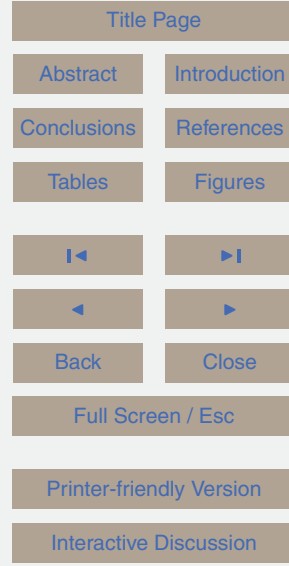

Title Page

Abstract · Introduction

Conclusions · References

Tables · Figures

|◄ · ►|

◄ · ►

Back · Close

**NHESSD**

doi:10.5194/nhess-2015-333

**Ground motion and threshold values for colluvium slope displacement**

C.-J. Jeng and D.-Z. Sue

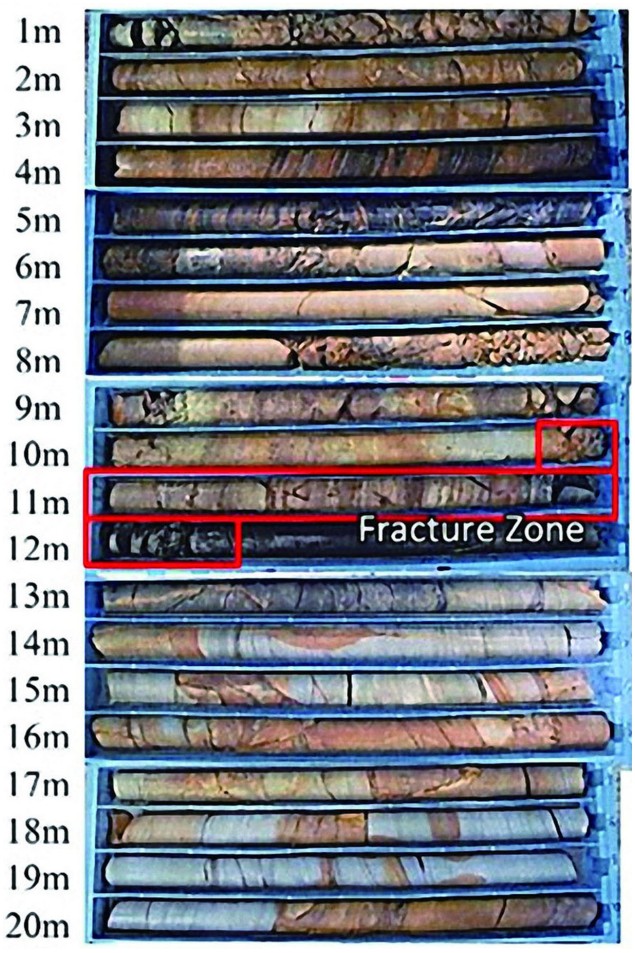

**Figure 13.** Rock core of borehole No. SIS-14. The rock core was collected when the inclinometer hole was drilled.

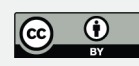

Discussion Paper | Discussion Paper | Discussion Paper | Discussion Paper | Discussion Paper

**NHESSD**

doi:10.5194/nhess-2015-333

**Ground motion and threshold values for colluvium slope displacement**

C.-J. Jeng and D.-Z. Sue

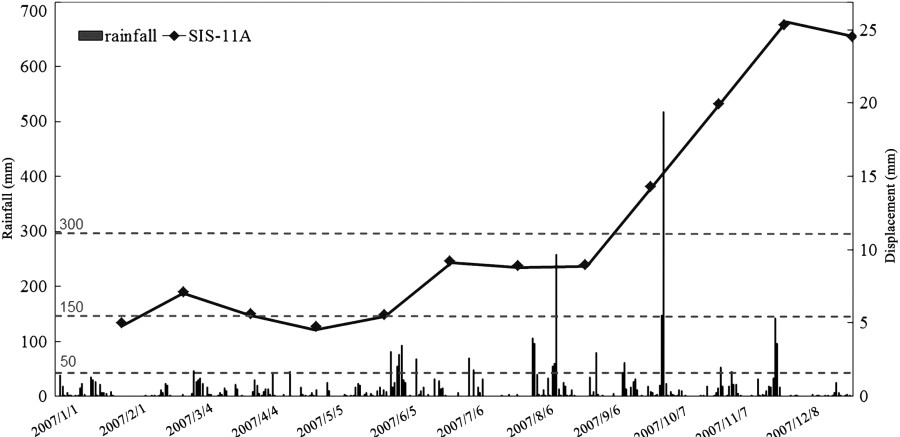

**Figure 14.** Relationship between displacement measured by the inclinometer (No. SIS-11A) and daily rainfall.

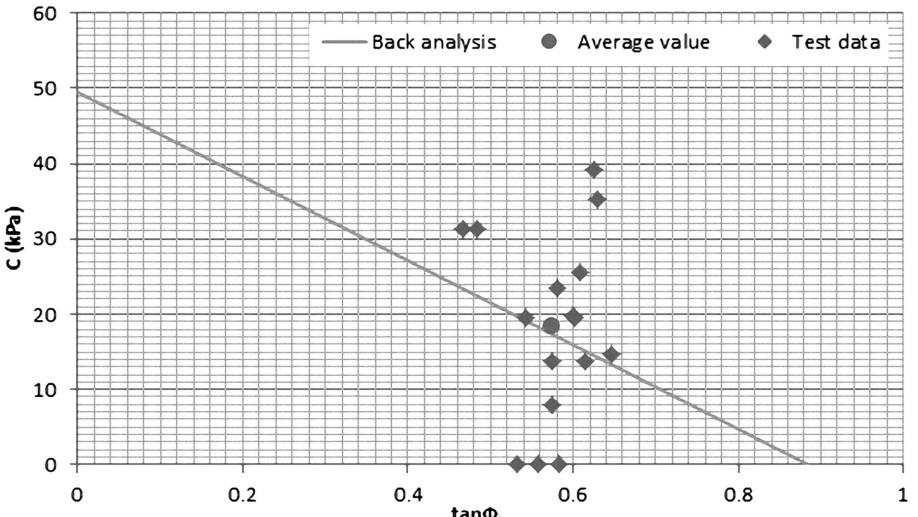

**Figure 15.** Back analysis of soil parameters for the colluvium layer. The cohesion and friction angle of the colluvium soil adopted were $C = 18.5$ kPa and $\phi = 29.6°$, respectively.

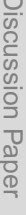

# NHESSD

doi:10.5194/nhess-2015-333

**Ground motion and threshold values for colluvium slope displacement**

C.-J. Jeng and D.-Z. Sue

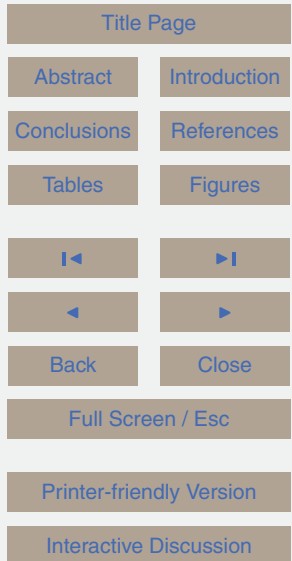

**NHESSD**

doi:10.5194/nhess-2015-333

**Ground motion and threshold values for colluvium slope displacement**

C.-J. Jeng and D.-Z. Sue

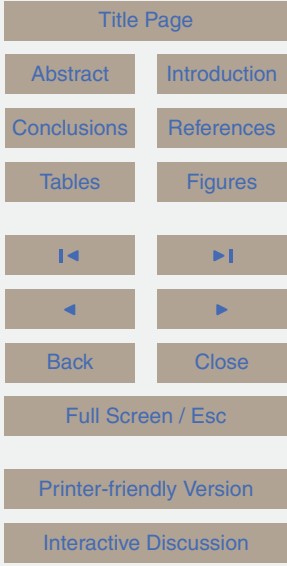

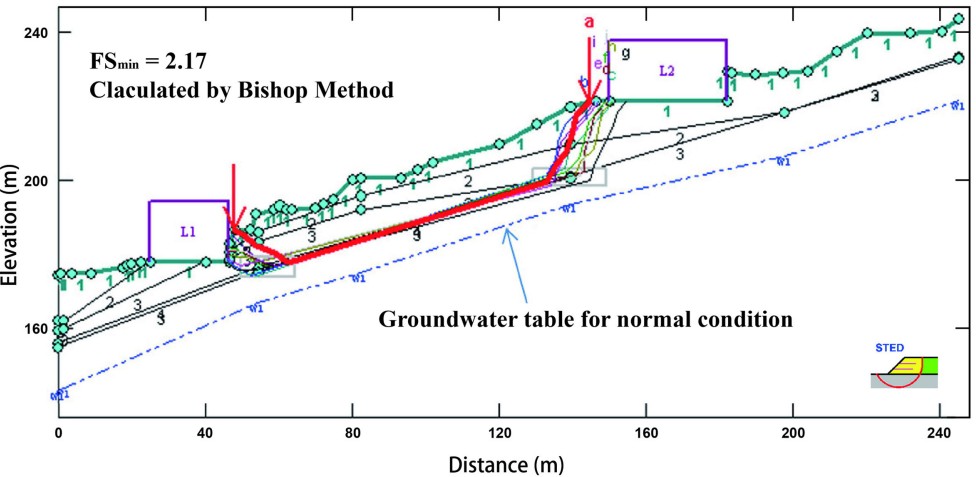

**Figure 16.** Slope stability for normal condition ($FS_{min}$ = 2.17).

Discussion Paper | Discussion Paper | Discussion Paper | Discussion Paper



**NHESSD**

doi:10.5194/nhess-2015-333

**Ground motion and threshold values for colluvium slope displacement**

C.-J. Jeng and D.-Z. Sue

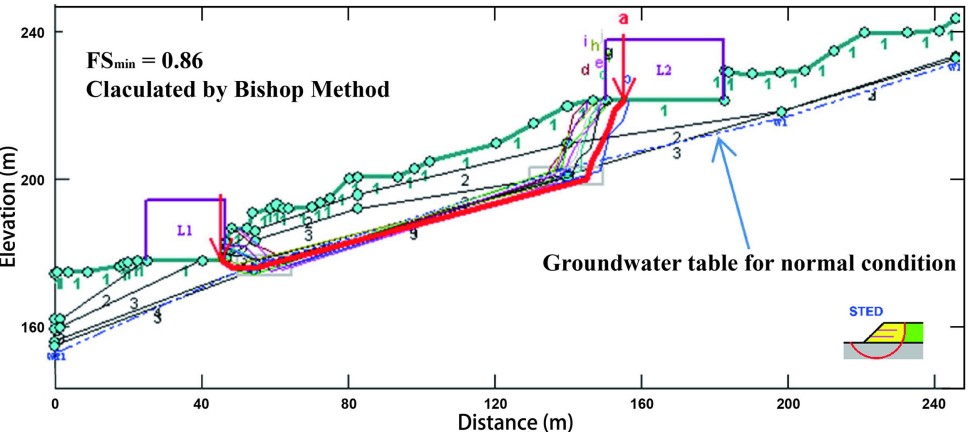

**Figure 17.** Slope stability for rainfall condition ($FS_{min}$ = 0.86).

Discussion Paper | Discussion Paper | Discussion Paper | Discussion Paper



**NHESSD**

doi:10.5194/nhess-2015-333

**Ground motion and threshold values for colluvium slope displacement**

C.-J. Jeng and D.-Z. Sue

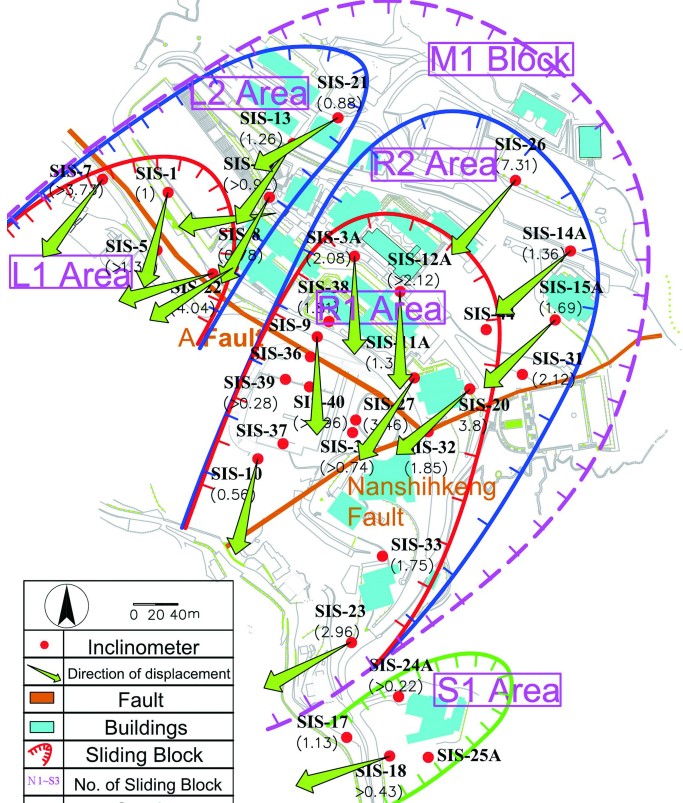

**Figure 18.** Sliding blocks with displacement directions within the campus. These results are derived from the inclinometer data and surface cracks observed over 2006 to 2008. Displacement rates in $mm\,month^{-1}$ are also shown in parentheses under the name of each inclinometer. The displacement rate, depth, and area of each block are important factors for slope stability analyses.

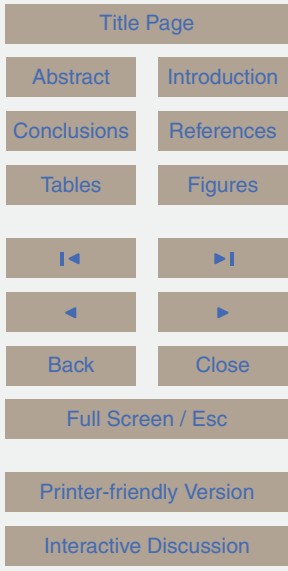

**Figure 19.** Flow directions of groundwater within the campus. The flow directions are inferred by the groundwater table drop, which is revealed by the contour of the groundwater table.

Discussion Paper | Discussion Paper | Discussion Paper | Discussion Paper |

**NHESSD**

doi:10.5194/nhess-2015-333

**Ground motion and threshold values for colluvium slope displacement**

C.-J. Jeng and D.-Z. Sue

# NHESSD

doi:10.5194/nhess-2015-333

**Ground motion and threshold values for colluvium slope displacement**

C.-J. Jeng and D.-Z. Sue

Title Page

Abstract · Introduction

Conclusions · References

Tables · Figures

|◄ · ►|

◄ · ►

Back · Close



M1 Block

L2 Area
B4

B8

L1 Area

**A-2**

R2 Area

**A-3**

R1 Area

**A-4**

B1  B3

**A-1**

B5

B2

B6

S1 Area

**A-5**

Previous estimated sliding blocks

New estimated sliding blocks

Cracks

0 20 40m

**Figure 20.** Sliding blocks estimated based on displacement and settlement of ground surface marks.

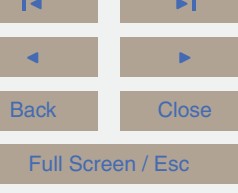

**Figure 21.** Locations of the catchpits within the campus.

**NHESSD**

doi:10.5194/nhess-2015-333

**Ground motion and threshold values for colluvium slope displacement**

C.-J. Jeng and D.-Z. Sue

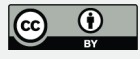

**NHESSD**

doi:10.5194/nhess-2015-333

**Ground motion and threshold values for colluvium slope displacement**

C.-J. Jeng and D.-Z. Sue

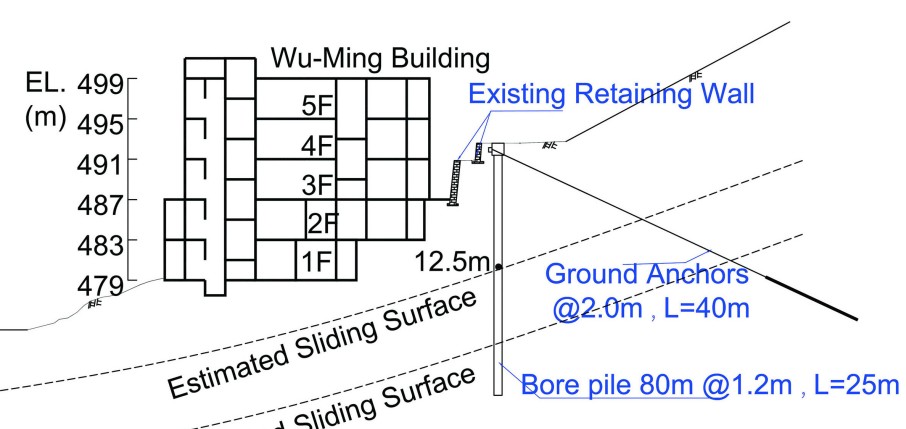

**Figure 22.** Strengthening construction for the Wu-Ming Building (B2) via bored piles with tieback anchors.

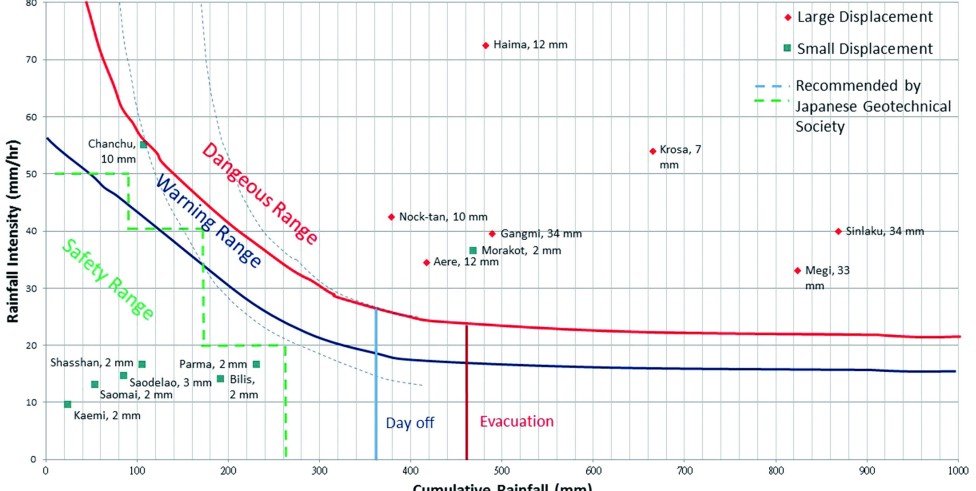

**Figure 23.** Threshold value curves (red and dark blue) correlated with slope displacement. The slope displacement (diamond and square marks) induced by typhoons over 2003 to 2010 can be derived from the relationship between average rainfall intensity and cumulative rainfall. Displacement less than 2 and over 10 mm is accordingly defined as threshold values to distinguish three ranges of safe, warning, and dangerous. Dashed curves recommended by the Japanese Geotechnical Society are also plotted for comparison.

Discussion Paper | Discussion Paper | Discussion Paper | Discussion Paper

**NHESSD**

doi:10.5194/nhess-2015-333

**Ground motion and threshold values for colluvium slope displacement**

C.-J. Jeng and D.-Z. Sue