# Peer review of "Characteristics of ground motion and threshold values for colluvium slope displacement induced by heavy rainfall: a case study in northern Taiwan"

_Natural Hazards and Earth System Sciences, 2015_

## Referee Comment (RC1) · Anonymous Referee #1 · 23 Jan 2016

For slope displacements occurred in such a case involved complexities of topography and geologic conditions, displacement analysis could be conducted using the 2D or 3D FEM programs. It would be nice to see some comparisons on the monitoring data with FEM predictions. Despite that, this content of this paper is quite interesting and well presented. It provides some useful information for slope engineering works.

---

## Short Comment (SC1) · 23 Feb 2016

This paper reveals valuable data of monitoring the dip-slope over years, and based on the data, correlation between slope movement and precipitation is further confirmed. The authors also present a contribution of warning values for the slope displacement on the purpose of mitigation of disaster. Although the warning values are relatively localized, for the geological conditions differ from other places, they can still be good references of risk assessment for cases with similar failure mechanisms.

---

## Referee Comment (RC2) · Anonymous Referee #2 · 24 Feb 2016

The paper studies the effects of heavy rainfall on the colluvium slopes in northeastern Taiwan. The significance of the settlement and displacement of the slopes recorded by an impressive number of devices (295 monitoring marks for ground motion and over thirty inclinometers) are evaluated. Threshold value curves that consider the displacement of slopes due to typhoon rainfall are established. The text is written clearly and it is properly organized. All figures and tables included in the text are necessary and appropriate. The abstract accurately reflects the contents. The research is original and is interesting because it presents a very good example of analysis of data coming from an high number of long-term monitoring ground devices in a complex topographical

and geological context. I would like to make only one remark. In Fig.15 the test data highlight an high variation of the soil cohesion (from 0 to 40 kPa) of the colluvial soils. Nevertheless, the authors adopted for the cohesion an average value of C = 18.5 kPa. Have the authors done a sensitivity analysis for the parameter C in the stability analysis? Which is the spatial variability of this parameter? Could the authors make some comments about that? Minor comments: Fig.1 please replace "altitudes" with "bed attitudes" Fig.17: I think that there is a mistake because the figure shows the groundwater table for normal condition and not in storm condition.

---

## Author Comment (AC1) · 27 Feb 2016

We would like to thank the Referee for the valuable comments and suggestions on this research. In this study, we focus on the topic of correlation between precipitation and slope stability in the background of rainy climate of Taiwan, and accordingly propose the modified warning values of predicting slope stability for the purpose of slope disaster prevention. As for the application of the Finite Element Method (FEM) to analyses of slope displacement raised by the Referee, in fact, we do have done a series of simulations using the FEM analysis method in 2D. Results and discussions of the simulations

are to be presented in another paper, which has been in preparation.

---

## Author Comment (AC2) · 27 Feb 2016

Dear Mr./Ms. Tseng, We wish to thank you for your worthy reviews and comments. Our research is actually a case study but we do apply various instruments to monitor the slope on which the entire campus is located. Through this case, we integrated all the data derived from the monitoring system to propose the warning values of predicting slope stability. This proves practicality of connection among the observation data and information of the monitoring system. Hence we hope that our study case can provide a useful and practical reference for the future cases which have similarities in geological

and climatic conditions.

---

## Author Comment (AC3) · 29 Feb 2016

We would like to thank the Referee for the thoughtful comments and suggestions on this paper. Firstly, concerning the question about the data shown in Figure 15, we obtained a standard deviation of the C values derived from the 16 colluvium soil samples as to be 12.04. This high standard deviation of the C values may be owing to widespread sampling of the colluvium soil within the campus. These samples of colluvium soil may possess different weathering degrees and different lithological components, causing different grain-size distribution and mechanic characteristics, for erosional processes

including soil erosion and mass wasting may lead to heterogeneity in nature of the colluvium soil within the campus. These interpretations and the standard deviation value will also be added in the text in the future version of this paper. As for the typos in Figures 1 and 17, revisions will be made in the future version.
* * *